# Modeling of Rapid Pam Systems Based on Electrothermal Micromirror for High-Resolution Facial Angiography

**DOI:** 10.3390/s23052592

**Published:** 2023-02-26

**Authors:** Yuanlin Xia, Yujie Wang, Tianxiang Liang, Zhen Peng, Liang He, Zhuqing Wang

**Affiliations:** School of Mechanical Engineering, Sichuan University, Chengdu 610065, China

**Keywords:** photoacoustic microscopy, MEMS, electrothermal micromirror, portable

## Abstract

In this paper, a portable photoacoustic microscopy (PAM) system is proposed based on a large stroke electrothermal micromirror to achieve high resolution and fast imaging. The crucial micromirror in the system realizes a precise and efficient 2-axis control. Two different designs of electrothermal actuators with “O” and “Z” shape are evenly located around the four directions of mirror plate. With a symmetrical structure, the actuator realized single direction drive only. The finite element modelling of both two proposed micromirror has realized a large displacement over 550 μm and the scan angle over ±30.43° at 0–10 V DC excitation. In addition, the steady-state and transient-state response show a high linearity and quick response respectively, which can contribute to a fast and stable imaging. Using the Linescan model, the system achieves an effective imaging area of 1 mm × 3 mm in 14 s and 1 mm × 4 mm in 12 s for the “O” and “Z” types, respectively. The proposed PAM systems have advantages in image resolution and control accuracy, indicating a significant potential in the field of facial angiography.

## 1. Introduction

Photoacoustic imaging (PAI) uses lasers with an ultrashort pulse duration to irradiate biological tissues. After the biological tissues absorb the laser, they are thermally expanded and dissipated by high-frequency vibration resulting in generating ultrasonic waves, as shown in Figure 1. Since Theodore Bowen et al. achieved photoacoustic imaging of soft tissues in 1981 [1], PAI is widely studied, especially for medical imaging related applications. With high contrast, non-contact imaging, high resolution and high signal noise ratio (SNR), the current applications of photoacoustic imaging include but not limited to, vascular analysis [2,3,4], oncology [5,6], neuroscience [7,8,9], ophthalmology [10], dermatology [11,12,13], gastroenterology [14], cardiology [15].

Photoacoustic microscopy (PAM) is a kind of PAI with high resolution which is achieved through scanning point by point, including optical scanning and acoustic scanning. Optical resolution photoacoustic microscopy imaging (OR-PAM) is one of the fastest growing PAM technologies, which can reach sub-micron resolution [16,17,18,19]. Desktop OR-PAM systems typically employ motors to drive laser heads to achieve point-by-point scanning. To ensure the imaging accuracy, the image acquisition speed is slow, making the desktop OR-PAM system susceptible to motion artifacts. At the same time, the use of a mechanical scanner also makes the system a large volume, which limits its application scenarios, as well as being difficult to study transient physiological processes [20]. To solve the above problems, many teams have started related research and found that scanning method was the key to high-speed imaging and equipment miniaturization [16,17,18,19,21]. Maslov et al. used a high-repetition rate pulsed laser combined with a synchronously controlled data acquisition system to achieve high-speed imaging of photoacoustic microscopy [22]. Qi et al. realized a fast PAM system using MEMS micromirror [23]. The use of high repetition rate lasers requires careful consideration of the power to prevent harm to the tissue [22]. MEMS micromirror as a key component in the novel laser scanning system can effectively reduce the system volume and improve the imaging speed and accuracy. Therefore, the miniaturization and rapid photoacoustic microscopy imaging equipment realized by MEMS technology is more in line with the development trend.

MEMS micromirror is a multifunctional active optical device [24]. According to different driving principles, it can be divided into electromagnetic, piezoelectric, electrostatic and electrothermal [25]. Electromagnetic micromirrors and electrostatic micromirrors usually require a large driving voltage. Piezoelectric scanning micromirrors are limited by the piezoelectric effect and have a small deflection angle. The electrothermal type can achieve a large displacement range under low voltage and is low-cost and easy to manufacture. It is widely used in medical laser scanning imaging equipment [26]. Todd et al. proposed a bimorph electrothermal actuator [27], and Jia et al. proposed a MEMS scanning galvanometer with an axial displacement of 480μm and a deflection angle of 30° based on a similar bimorph electrothermal actuator [28].

In this research, we propose a PAM system based on electrothermal micromirror. The micromirror has a mirror diameter of 1000 μm. Two different designs of electrothermal actuators with “O” and “Z” shape are evenly located around the four directions of mirror plate, which can achieve a large displacement and deflection angle. FEM models of the two types micromirror in room temperature (293.15 K) have been built to demonstrate the reflectivity and piston thermal actuation of the micromirror. The imaging simulations were also conducted to investigate the imaging area that the portable PAM system may achieve.

## 2. Materials and Methods

### 2.1. The Design of PAM System

Figure 2 is a schematic of the elements in the portable OR-PAM system with the size smaller than Φ10 cm × 10 cm. A laser beam is directed into the system through a collimator. The collimated laser is converged by a focus lens and re-directed to a 2-axis micromirror with an adjusted reflector to act on the tissue through a cover glass. Based on the photoacoustic effect, the tissue generates ultrasonic waves which are reflected by the cover glass to be received by the ultrasonic transducer. And the collected signal is used for subsequent image reconstruction. It is interesting to note that water is filled in the propagation path of the ultrasonic waves generated by tissues to ensure an efficient coupling for longitudinal waves, which will provide a better SNR [29,30].

### 2.2. Electrothermal Micromirror

#### 2.2.1. The Design of Micromirror

Referring to the structure proposed by Jia et al. [28], the electrothermal micromirror employs a biaxial gimbals structure allowing two-dimensional scanning of the mirror, which is the key component in the system to achieve the quick response and high resolution as well as. Two different types of electrothermal actuators with “O” and “Z” shape were designed. In Figure 3a, four electrothermal actuators with “O” shape are evenly located on the four sides of the central mirror plate. The mirror with a diameter of 1000 μm has to realize two-dimensional scanning and piston motion. The endpoints of two shafts are marked out in Figure 3a. The actuators are driven by joule heat, which will inevitably transfer to the mirror. Thus, to prevent the temperature difference between the center and the edge of the mirror, which will result in an unwanted deformation, the mirror plate is designed to be circular. In addition, a layer of SiO_2_ surrounds the mirror as the heat insulation to reduce the thermal deformation of the mirror itself. The electrothermal actuators with “Z” shape are shown in Figure 3b. It is worth noting that the mirror shape is designed as a regular octagon (“Z” shape) instead of the traditional rectangle or circle (“O” shape). As a rectangle, when the micromirror is working, the four corners of the rectangle are farther from the center of the mirror surface than other points, which is easy to cause uneven temperature distribution, causing the mirror surface itself to deform and serious scanning error. Moreover, considering that the circular shape is more troublesome to process in the MEMS process, the mirror surface may appear jagged at the edge. Therefore, in this work, for a clear distinction with “O” shape mirror, the surface plate of the “Z” shape micromirror is designed to be a regular octagon.

#### 2.2.2. The Electrothermal Actuator

Figure 4a is an electrothermal actuator of “O” type, mainly consisting of Al and SiO_2_, which utilizes the difference in the coefficient of thermal expansion (CTE) between two materials for operation. The fixed end is connected to the substrate, while the free end is connected to the mirror plate to provide movement. Figure 4b shows each layer of the electrothermal actuator and the related size information. The temperature of the beams can be changed by applying a voltage to the Ti resistors embedded along the actuator. The Ti resistor is electrically isolated from Al layer by thin layers of SiO_2_ whose thickness is much less than 1% of the entire actuator, thus it does not take the isolated layer into the model analysis. With the Al/SiO_2_ thickness ratio of 0.91, the device operating stability and reliability of the actuator increased [28]. The finite element method (FEM) simulation of the actuator in room temperature (293.15 K) is shown in Figure 4c. The convection heat transfer coefficient is set as 1750 W/(m^2^·K) according to the classical Morgan relation [31]. The simulation results of displacements in x, y and z axis show that the symmetrical structure can limit the displacement of x and y axis, while maintain a large displacement of z axis. The 3D model of the thermal actuator with “Z” type is shown in Figure 4d, whose fixed end is connected to the MEMS substrate to keep the actuator stable during operation. There are two electrodes extending from the inside of the actuator on the fixed end platform, which are the excitation signal and the ground signal respectively. The electrothermal actuator can be driven by applying the excitation signal at the electrodes. The whole electrothermal actuator is composed of SiO_2_ and Al alternately bonded up and down from the fixed end to the free end, thus it can be composed of 6 basic structures of bimorph electrothermal actuators or 2 S-shaped inverted-series-connected (ISC) structures.

Figure 4e shows the main structure and size information of the actuator, where the 2nd layer-Pb provides heat source as the thermal resistance. It is important to note that in this model there is no insulating layer between the 2nd layer-Pb and the 3rd layer-Al. Although the insulation layers are necessary in actual fabrication, the extremely thin insulation layers are not the dominant contribution to the actual deformation and will therefore not be addressed in the simulation. It is worth pointing out that in the actuator designed of this “Z” type, the thickness of SiO_2_ is 1.8 μm, the thickness of Al is 1.6 μm, and Al/SiO_2_ = 0.91, which is the same with “O” type. Performance test of the actuator in Figure 4f shows that with a 5 V excitation voltage, the free end of the electrothermal actuator produces a downward displacement in z axis, while there is nearly no displacement in another 2 axes. The test results indicated that the design meets the goal of suppressing irrelevant displacements and suit the basic characteristics of electrothermal actuator of low voltage leading to large displacement.

#### 2.2.3. Finite Element Simulation of Micromirror

A FEM model of the micromirror in room temperature (293.15 K) has been built to demonstrate the reflection and piston thermal actuation of the micromirror. To simplify the model, thin layers of SiO_2_ between Al and Ti were not included in the simulation. The simulation results of the “O” type and “Z” type are shown in Figure 5a–c and Figure 5d–f, respectively. As shown in Figure 5a, the mirror plate was elevated 331.89 μm out of plane with prestress of 400M Pa applied on the actuators. The prestress is caused by the residual stresses due to the fabrication process. The upward deformation induced by the prestress causes a large displacement of the mirror plate towards the downward direction, which means that the tilting angle of the mirror plate becomes larger than that of the undeformed structure. In Figure 5b, 10 V direct excitation was applied to all four actuators, resulting in a 463.53 μm downward motion of the mirror plate. The rotation can be realized by applying differential excitations to one of the two opposite actuators, while the other two actuators maintain their original states. In Figure 5c, only one actuator was applied 10 V excitation and the others maintained their states, which results in a 30.43° rotation of the mirror plate. The FEM results proved the micromirror can realize large stroke and wide adjustment range of the scanning angle.

The micromirror of the “Z” type has an axial displacement of about 653 μm away from the substrate in the initial state, as shown in Figure 5d. The piston motion state and deflection motion state of the micromirror are shown in Figure 5e,f, respectively. When only one actuator works, the micromirror is deflected towards the substrate on one side of the actuator and can achieve about 35° at 9 V. When the four electrothermal actuators are added the same excitation at the same time, the micromirror will be driven toward the substrate and can achieve a net displacement of 553 μm at 6 V.

### 2.3. Scanning

The scanning process of the PAM system can be simplified to the laser being reflected onto the target plane by the micromirror. The performed laser scanning was in the Linescan model, whose resolution largely depended on the frequency difference of rotation shafts and the performance of ultrasonic transducer [32], as shown in Figure 6. A higher-sampling-frequency ultrasonic transducer can shorten the gap of two sampling points, and the larger rotation frequency difference can smooth the scanning lines and make them denser. Considering that resolution actually refers to the distance between pixels, the denser scanning lines will provide a higher resolution image.

The whole scanning lines exhibits a feature of getting denser intensities from the edge to the middle, which results in a resolution depending on the edge region. Therefore, to ensure the image quality, the final image intercepts the central area of the original scanning area, called the effective imaging area. It is clear that the resolution of the effective imaging area should be determined by the scanning point at the edges.

## 3. Results

### 3.1. Stable State Response

The stable state response of the “O” type and “Z” type are shown in Figure 7a–d. The steady test results “O” type in Figure 7a demonstrate a displacement of 797.42 μm in the piston motion and a rotational angle of 30.43° at an 10 V DC. The piston motion is realized by driving all actuators with the same excitation at the same time, while the rotational motion is achieved by driving only one actuator and keeping the rest actuators in their initial states. Meanwhile, it can be confirmed in Figure 7b that there is an obvious linear relationship between voltage and the displacement as well as the rotation angle when the voltage is above 4.8 V. The goodness-of-fit values for voltage-displacement and voltage-angle are 0.9907 and 0.9869, respectively, showing accurate control of piston motion and a wide of rotation angles. The steady test results of “Z” type in Figure 7c demonstrate a displacement of about 553 μm in the piston motion and a rotational angle of about 35° at 6 V and 9 V DC, respectively. There is also an obvious linear relationship between voltage and displacement as well as the rotation angle for voltages are between 2.4 V to 6 V in piston motion and 2.4 V to 8 V in rotational motion.

### 3.2. Transient Response

From the transient curves at 10 V DC in Figure 8a,b, the tendency of transient response is similar between the “O” type and “Z” type, while the rise time of displacement and angle is a little different. The excitation method is performed in the same way as in the steady-state test. When keeping an excitation of 10 V, the mirror plate is actuated to a corresponding state and stabilizes itself. After the excitation vanishes, the mirror plate immediately returns to its initial state, which demonstrates the good manipulability of the micromirror. The short rise time and downtime also indicate a fast response to the excitation.

### 3.3. Imaging

#### 3.3.1. Driving Method

To find out whether the motion of the mirror plate under the excitation can be realized by Lissajous or Linescan scanning. Moreover, we also tried to figure out what is the difference in the response frequencies of the two designs at similar excitation amplitudes. To ensure quality of the laser scan, the excitations applied to the actuators are greatly different in the frequency. For the “O” type, the pair of actuators acting on the fast shaft are driven by the excitation of the 40 Hz AC with the amplitude of 10 V, while those of the slow shaft are driven by the excitation of the 0.5 Hz AC with the same amplitude. Thanks to the excellent fast response of the designed mirrors, the motion of the fast and slow axes is effectively fitted as a sinusoidal function for both types. As a result of the “O” type, the frequency of the fast shaft and the slow shaft is 80 Hz and 1 Hz, respectively, as shown in Figure 9a. In the two-dimensional plane, each motion around a shaft will control the scanning of the laser in the corresponding dimension. With such a large difference in the motional frequency between the fast and slow shaft, the scan route of the laser can be divided into a fast dimension and a slow dimension, which will result in a higher image resolution because of the denser scanning lines. As for the “Z” type, the fast and slow shaft are driven by the excitation of the 60 Hz and 0.2 Hz AC with the amplitude of 10 V, respectively. The motion of the fast axis and the slow axis is also effectively fitted to a sinusoidal function. The result shows that the rotation frequency of the fast axis is about 120 Hz, and the rotation frequency of the slow axis is about 0.4 Hz, as shown in Figure 9b.

#### 3.3.2. Imaging Result

Based on the results of the two-dimensional deflection scanning of the mirror in Section 3.3.1, the imaging simulation was conducted using the software MATLAB. The distance from the surface of the imaging area to the mirror plate is set to 4 mm. The simulation of system imaging is achieved by combining the theory of trigonometric functions with a dataset of time-dependent deflection angles. The imaging simulation of the “O” type shows that the portable PAM system achieves an effective imaging area of 1 mm × 3 mm in 14 s. From the scan lines at 2 s, 8 s, and 14 s shown in Figure 10a, the scan area expands from the center to the edges and the sparse and dense distribution of scan lines fits the Linescan model. Cropping a 1 mm × 3 mm area in the center of the scan line at 14 s as the effective imaging area and its lower right corner has the sparsest scan lines, as shown in Figure 10b. The largest gap of the scan lines is about 9 μm; thus, the resolution of the effective imaging area should be better than 9 μm. Notably, it is assumed here that the ultrasonic transducer has sufficient performance. Meanwhile, Figure 10c shows a PAM simulation of a blood vessel sample. When being irradiated by a laser, the PA signal sent by the blood vessel is received by ultrasonic transducer to reconstruct the tissue structure. As shown in Figure 10, the portable PAM system enables the fast and high-resolution images and can achieve the images of tiny tissues such as capillaries.

For the “Z” type, the scanning conditions of the system at 2 s, 7 s, and 12 s are described, respectively, as shown in Figure 11a. Clearly, the laser scanning is performed in a progressive scanning manner, which conforms to the Linescan mode. The density of the scan lines increases with time during the scan. At the same time, it can be seen from the scanning state at 2 s that during the scanning process, the scanning lines will overlap to a large extent in the central part, thus the resolution in the central area will be higher. According to the analysis of the line scan model above, the actual situation of imaging has been taken, and a 1 mm × 4 mm area is taken as the effective imaging area in the center of the image scanned at 12 s.

As shown in Figure 11b, to determine the resolution of the effective imaging area, the lower right corner area is selected to determine the resolution. The scan lines are nearly horizontal due to the large frequency difference in the deflection of the micromirror around the two axes of rotation. Analysis of the distance between the two scan lines in the image shows that the effective imaging area has a resolution of about 12 μm, which satisfies the design index of this project, and the resolution is better than 20 μm. An example is in Figure 11c.

The proposed two types of the large stroke electrothermal micromirror show a larger displacement and rotation angle as compared to the reported work [28]. Meanwhile, the fast response and small size show the potential aiming at the application of a PAM system. However, with the limitation of the resolution 9–12 μm may not be enough to do an angiography of smaller arteries (arterioles), smaller veins (venules) and capillaries. At present, only venules and arterioles with diameter around 100 μm should be clearly visible in the PAM generated images. Moreover, as compared with traditional magnetic resonance imaging (MRI) and computed tomography (CT) methods which are likely to be a lot more expensive, time-consuming, worse resolution, PAM provides a much more cost-effective and high-resolution method. However, MRI and CT are able to investigate deeper blood vessels than the PAM would be able to in the current design. Further optimization for higher resolution and deeper detection is beneficial to incorporate the context of the truly impressive resolution achieved by the present design into biological applications. What’s more, a future study that could be worth including and considering would be to study subcutaneous vessel patterns and distributions for further medical applications, and to test the performance of the PAM device with the new micromirror design in real patients with the aid of a clinical or basic medical scientist and potentially verify how fine the resolution is in terms of the smaller vessels [33,34,35].

## 4. Conclusions

This research introduces a portable PAM system based on the electrothermal micromirror and analyses the performance of the crucial element micromirror as well as the imaging process. Two different designs of electrothermal actuators with “O” and “Z” shape were designed. Both the two micromirrors replace the traditional motors to realize fast and high-precision laser scanning, as well as reducing the size of the PAM system. For the “O” type, the FEM model shows that the micromirror achieves a large displacement of about 797.42 μm and the scan angle up to ±30.43° at 10 V DC excitation. For the “Z” type, the micromirror achieves a large displacement of 553 μm and the scan angle of 35° at 6 V and 9 V DC excitation, respectively. Meanwhile, the steady-state and transient-state response of the micromirror indicate that the PAM system realizes a fast and stable imaging. With the excitations applied to the actuators greatly differing in the frequency, the system achieves an ideal Linescan process. The simulation result shows that the “O” type system achieves an effeactive imaging area of 1 mm × 3 mm in 14 s with a 9 μm resolution, while the “Z” type system achieves an effective imaging area of 1 mm × 4 mm in 12 s with a 12 μm resolution. The results above indicate that the proposed PAM systems have a great potential in portable imaging applications in various environments, such as outdoor facial angiography.

## Figures and Tables

**Figure 1 sensors-23-02592-f001:**
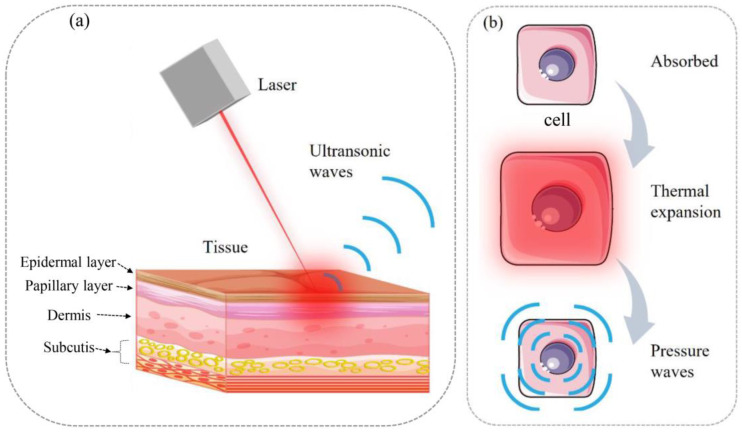
(**a**) Photoacoustic effect. (**b**) The change of tissues in microscale.

**Figure 2 sensors-23-02592-f002:**
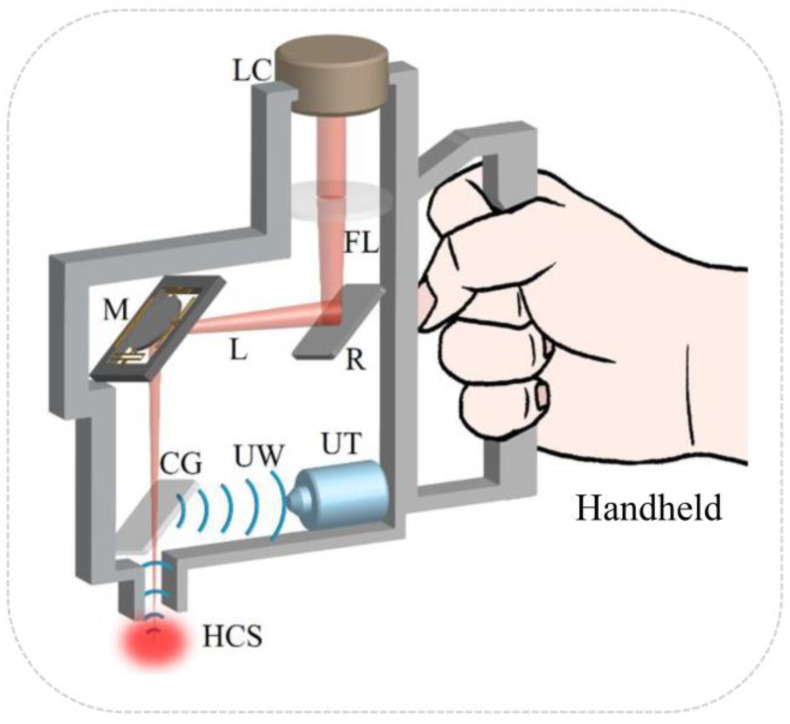
The portable OR-PAM system. LC, laser collimator; FL, focus lens; R, reflector; L, laser; M, micromirror; CG, cover glass; UW, ultrasonic wave; UT, ultrasonic transducer; HCS, highly converging spot.

**Figure 3 sensors-23-02592-f003:**
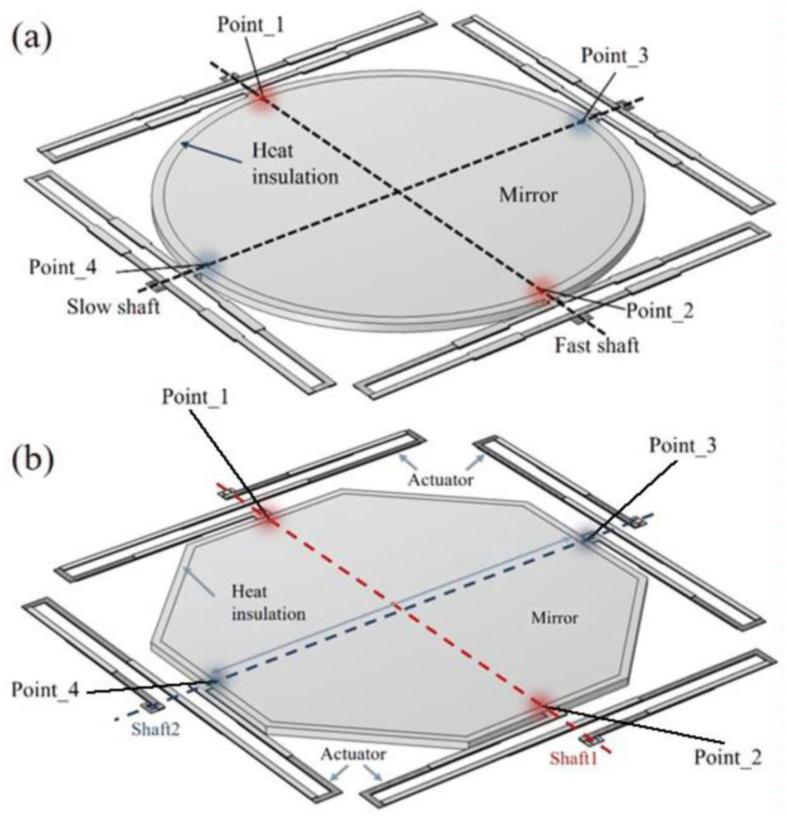
The micromirrors. (**a**) “O” type. (**b**) “Z” type.

**Figure 4 sensors-23-02592-f004:**
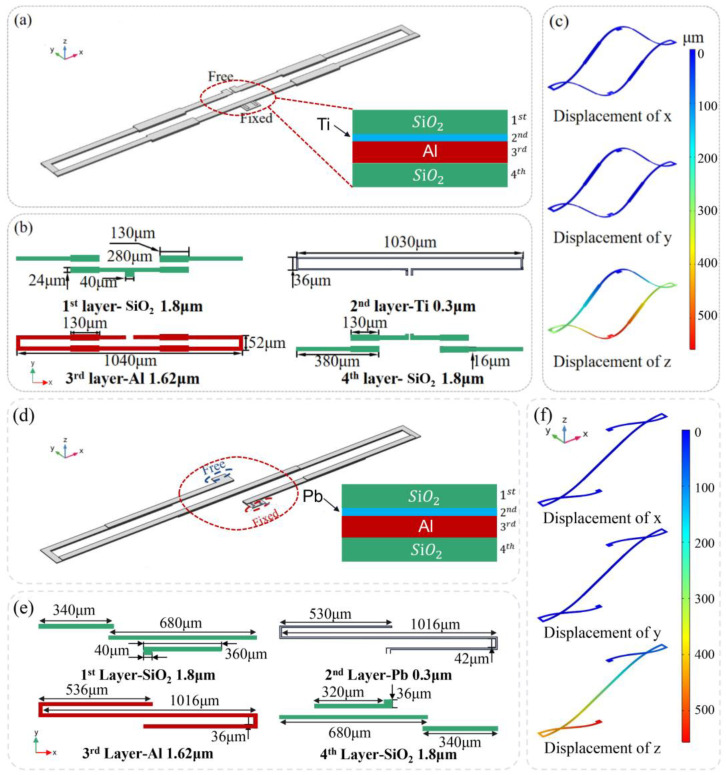
Structure of actuators. (**a**–**c**) “O” type. (**b**–**f**) “Z” type. (**a**,**d**) Electrothermal actuator. (**b**,**e**) Each layer of the actuator and the related size information. (**c**,**f**) The simulation of the actuator.

**Figure 5 sensors-23-02592-f005:**
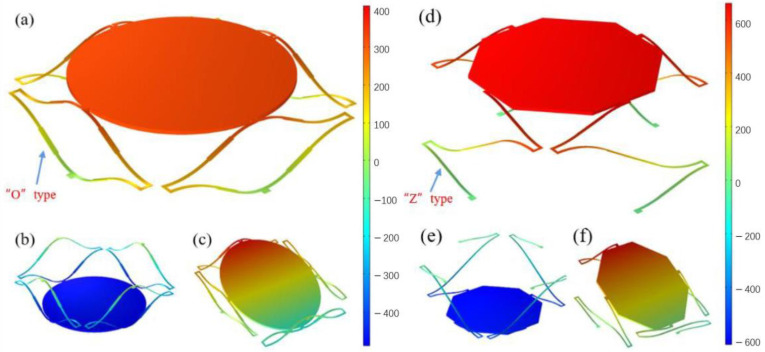
The FEM modeling of the micromirror. (**a**–**c**) “O” type. (**b**–**f**) “Z” type. (**a**,**d**) The initial elevation of the mirror plate due to the prestress. (**b**,**e**) The piston motion of the mirror plate when four actuators work in the same state. (**c**,**f**) The rotation motion of the mirror plate when only one actuator works.

**Figure 6 sensors-23-02592-f006:**
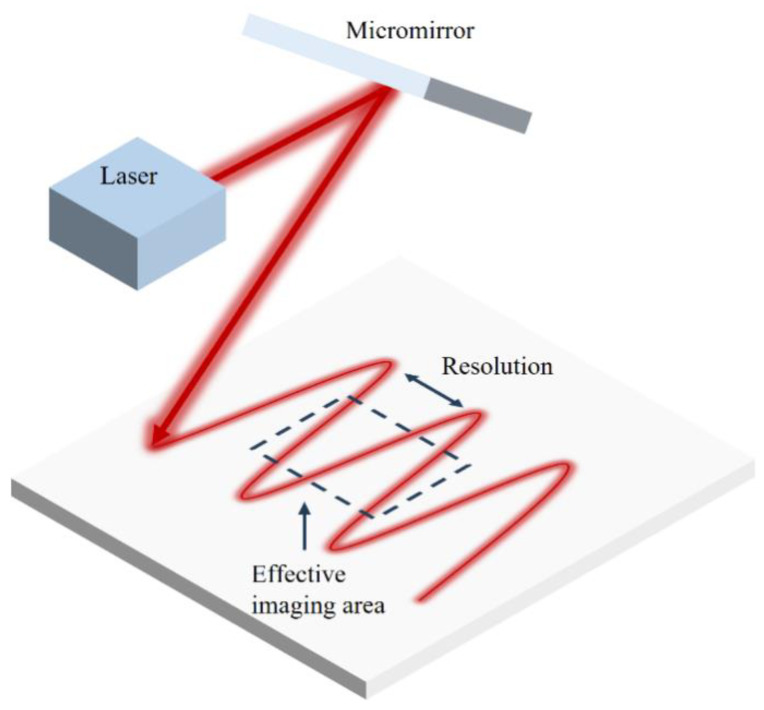
The schematic of Linescan mode.

**Figure 7 sensors-23-02592-f007:**
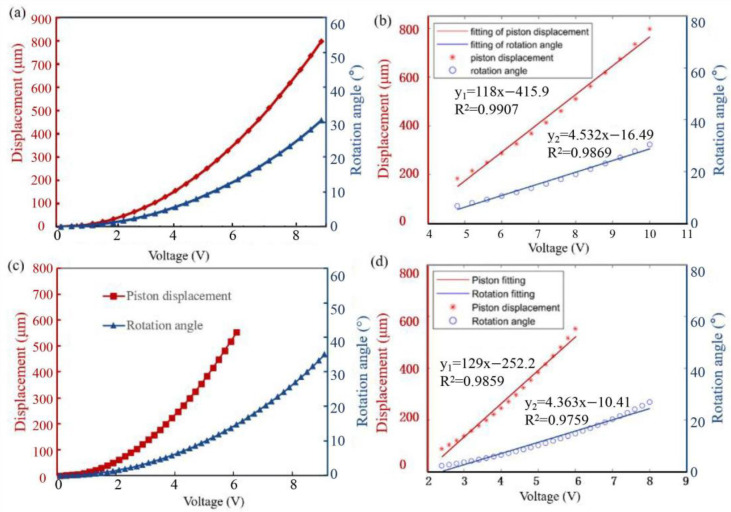
Stable state response. (**a**,**b**) “O” type. (**c**,**d**) “Z” type. (**a**,**c**) The steady curves. (**b**,**d**) Univariate Linear Fit to Data.

**Figure 8 sensors-23-02592-f008:**
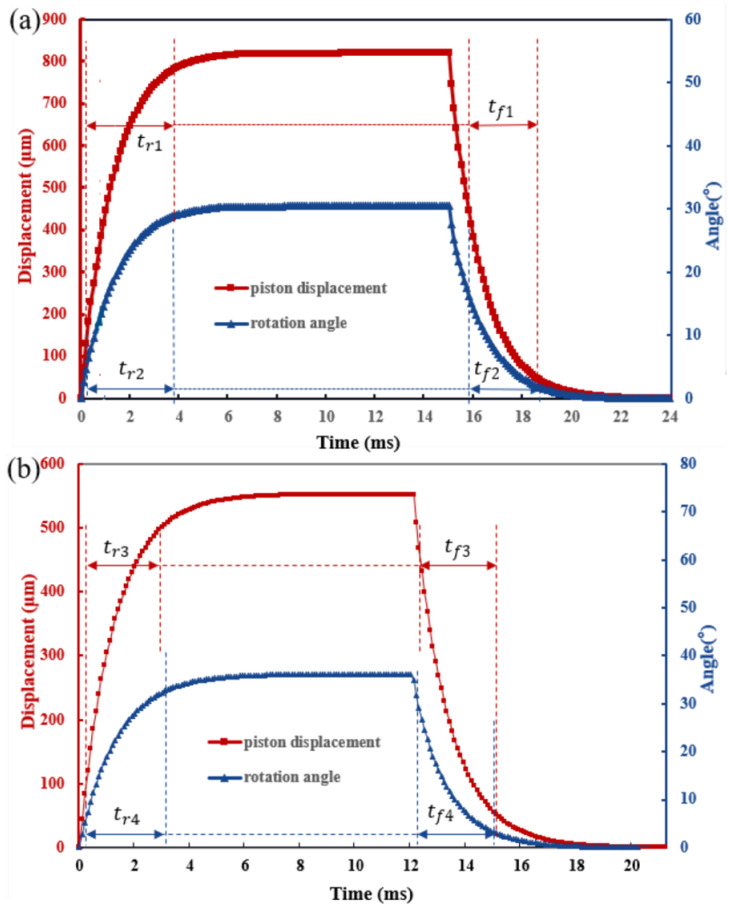
The transient curves. (**a**) “O” type. t**_r1_** = 3.7 ms, t**_r2_** = 3.6 ms, t**_f1_** = 2.8 ms, t**_f2_** = 3 ms. (**b**) “Z” type. t**_r_**_3_ = 2.9 ms, t**_r_**_4_ = 2.7 ms, t**_f_**_3_ = 2.8 ms, t**_f_**_4_ = 3 ms.

**Figure 9 sensors-23-02592-f009:**
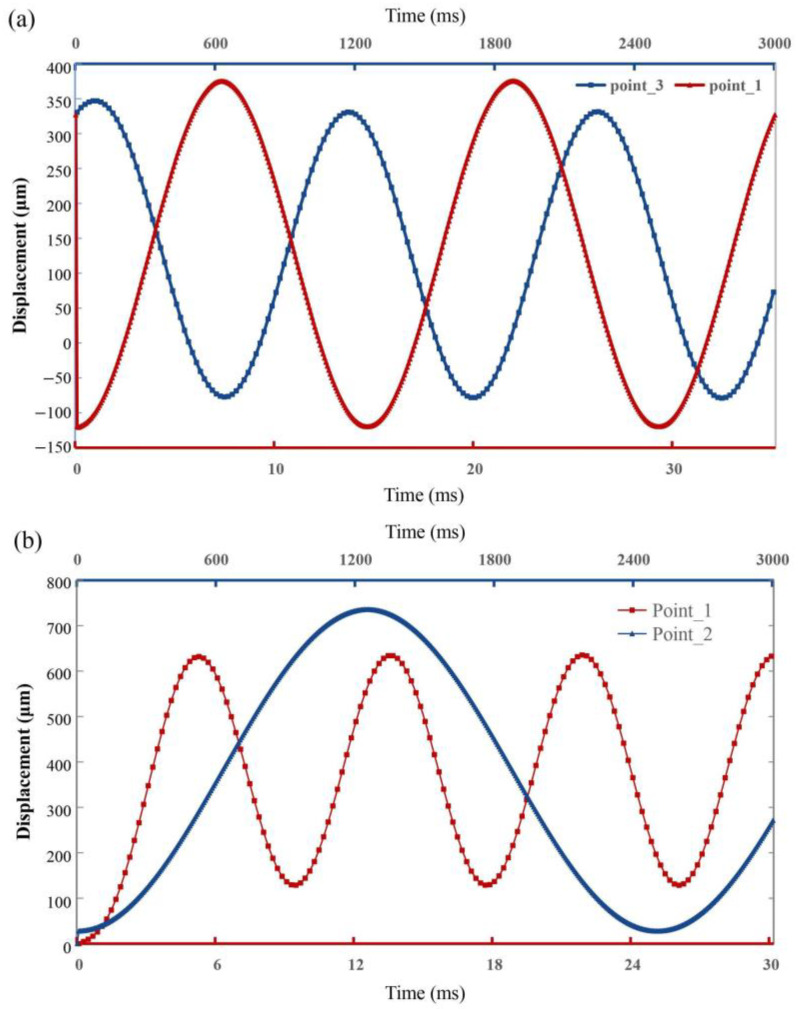
The frequencies of the fast shaft and the slow shaft. (**a**) “O” type. (**b**) “Z” type.

**Figure 10 sensors-23-02592-f010:**
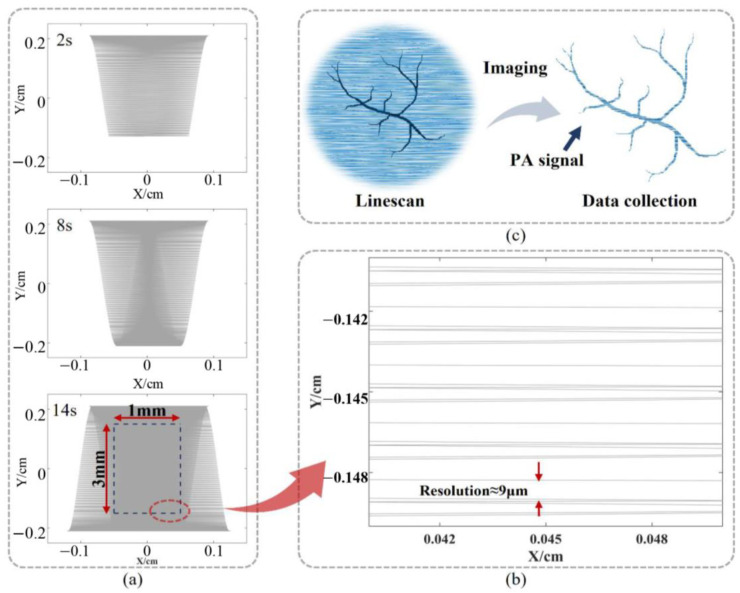
The imaging simulation of “O” type. (**a**) The scan state changes over time and results in an effective imaging area of 1 mm × 3 mm at 14 s. (**b**) The edge area of the effective imaging area, indicating the resolution of the image is better than 9 μm. (**c**) The image simulation of a blood vessel sample.

**Figure 11 sensors-23-02592-f011:**
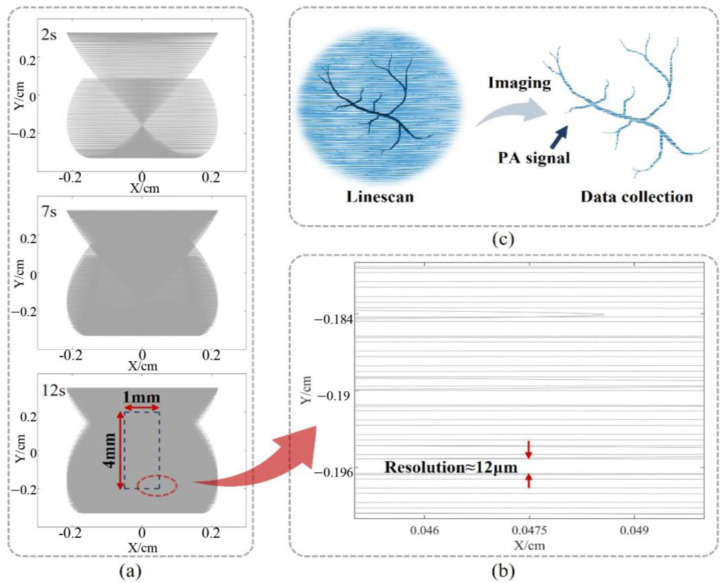
The imaging simulation of “Z” type. (**a**) The scan state changes over time and results in an effective imaging area of 1 mm × 3 mm at 14 s. (**b**) The edge area of the effective imaging area, indicating the resolution of the image is better than 9 μm. (**c**) The image simulation of a blood vessel sample.

## Data Availability

As there are further experiments based on this study, research data will not be shared for now.

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
