# Peer review of "Modeling of Rapid Pam Systems Based on Electrothermal Micromirror for High-Resolution Facial Angiography"

_sensors, 2023, doi:10.3390/s23052592_

Round 1
Reviewer 1 Report
This manuscript is a detailed breakdown and rationale of the design of a novel approach to photoacoustic microscopy using an ingeniously modelled electrothermal actuator controlled micromirror. Overall, the paper is well written and describes at an appropriate level for the technical audience of the Journal how the device and micromirror are designed and function (in the FEM setting) clearly.
This information is presented at a good level even for a non-engineering audience. As an anatomist myself, I was in fact able, with the aid of some additional readings on the basics of electrothermal actuators, to follow and comprehend how the device is designed and operates, thanks to the authors’ clear descriptions and fantastic schematics.
As an anatomist I am unable to comment further on the actual design and functionality of the device but find the micromirror and the actuator models of high quality and appropriate for their intended use, from my understanding of the paper.
The main suggestions I have for the manuscript include some small grammatical/syntax corrections and recommendations on integrating more information on the application of the device with some practical examples from similar devices or applications.
Please see list of recommendations below:
Grammar and Writing:
1. Line 72: please change “angel” to “angle”.
2. Line 98: please change “portable” to “portability”
3. Line 101: please change “two shafts of slow and fast respectively” to “a slow and fast shaft”
4. Line 105: please remove “a” so that the sentence reads: “the mirror plate is designed to be circular”
5. Line 127: please change “electric” to “electrically”
6. Line 130: please remove the word “get” so that the sentence reads: “ the device operating stability and reliability of the actuator increased”.
7. Line 140 and 148: please fix reference to Figure 4.
8. Line 151: please change sentence to say “The authors postulate that…” instead of “The project believes that...”
9. Line 160: Sentence is incomplete, please remove or include missing text.
10.
Content:
1. Line 157: If it is appropriate for the audience of the journal, please explain how the ratio and thickness of the SiO2 and Al components actuators enhance stability and reliability of the actuator structure overall. If it is immediately obvious to other mechanical engineers, please ignore this comment.
2. Figure 4: This schematic was very helpful to understand the actuator design. However, indicating where the horizontal sections through the actuators are visualized from in image 4e and 4d would be of benefit to the reader. Please include labels (A, B, C, and D) and section lines in Image 4a and 4d showing where each of the 4 layers of SiO2, Pb, and Al are located in the superior view of the actuators.
3. Line 183: Please reference the statement saying that prestress is believed to protect the structure. Unless this is common knowledge in the authors’ field, you may ignore this comment.
4. Results & discussion: depending on the primary application of the device, the very impressive resolution of 9-12μm the authors have achieved with it may not be enough to do an angiography of smaller arteries (arterioles), smaller veins (venules) and capillaries. To my knowledge, capillaries are usually between 4-10μm in width, meaning that they would not be visualized clearly even by the 9μm resolution. Venules and arterioles are around the 100μm diameter mark and should be clearly visible in the PAM generated images. This information can generally be found in histology and anatomy textbooks and can be of benefit to include the context of the truly impressive resolution achieved by the authors’ design into the biological applciation. Since the application is very much directed at medical diagnosis and angiography, I would strongly recommend that the authors briefly contextualize the findings of the study in its application either in the conclusion or immediately before it.
5. Discussion and conclusion: mention is made of application of this technology for outdoor facial angiography, but the details of its application and utility are not presented clearly. I believe that the paper could benefit by outlining for an audience in the clinical or basic medical sciences what the applications, potential benefits and downsides of this specific improved device design are. This could be achieved by presenting, in the conclusion, examples of applications from the literature, explaining how the improvement that the authors have made on the original PAM design improves on the resolution compared to other methods and other devices normally used for angiography. Especially since MRI and CT methods would likely be a lot more expensive, time consuming, and have worse resolution. However, they are able to investigate deeper blood vessels than the PAM would be able to in its current design. Research applications can also be discussed, where using this PAM device to study subcutaneous vessel patterns and distributions for further medical applications could be of use. From the medical literature some useful sources to discuss could be:
a. CT Angiography for Surgical Planning in Face Transplantation Candidates, Soga et al., 2013, Am. J Neuroradiol.; https://doi.org/10.3174%2Fajnr.A3268
b. Optical Coherence Tomography Angiograpyhy Using the Optovue Device, Huang et al, 2016, Dev. Opthalmol. Basel, Karger; https://doi.org/10.1159/000442770
c. Magnetic Resonance Angiography in Facial and other Pain: Neurovascular Mechanisms of Trigeminal Sensation, May et al., 2016, Journal of Cerebral Blood Flow and Metabolism; https://doi.org/10.1097/00004647-200110000-00005
6. A future study recommendation that could be worth including and considering for the authors would be to test the performance of the PAM device with the new micromirror design in real patients with the aid of a clinical or basic medical scientist and potentially verify how fine the resolution is in terms of the smaller vessels. The above paper by May et al 2016 (https://doi.org/10.1097/00004647-200110000-00005) could be of use in designing such a follow-up study.
Reviewer 2 Report
- You have to add a discussion paragraph to evaluate your results with previous work
- page 4, line 140 & page 5, line 148 &: Eliminate the 'n Error! Reference source not 140 found'
- Try to improve the quality of Figure 4.
Reviewer 3 Report
The authors proposed two types of electrothermal micromirrors for a PAM system to achieve high-resolution imaging. However, the manuscript is poorly written, and many mistakes are found throughout the paper. In addition, the lack of experimental data also prevents the readers from verifying the system's performance. It cannot be accepted before addressing the following comments and improving the scientific.
Here are some comments for improvements:
(1) The quality of all the figures should be improved.
(2) There are so many typos throughout the manuscript, including but not limited to:
- Error! Reference source not found in Line 140, 148.
- In line 154, what is the meaning of "the difference between the second layer of Pb and the third layer 154 of Al is Insulation layers are ignored"?
- The unit of voltage is sometimes lowercase and sometimes uppercase throughout the paper.
- In line 170, The expression of "weren't" is informal.
- In Figure 7, even the drawing tools in Chinese are shown.
- In the legend of Figure 8, "tr1" to "tf2" should be consistent with the figure.
I believe that there would be more mistakes in the manuscript. The authors must carefully check the manuscript and revise them.
(3) What is Figure 9 trying to explain? What does "time of point_1,2,3" represent? The authors should add more content to clarify the meaning of the figures.
(4) How do the authors do the imaging simulation? The authors should clarify it in the paper
(5) The imaging simulation shows good imaging performance. However, the authors mentioned that "we propose a PAM system based on electrothermal micromirror.". Therefore, experimental imaging results should also be shown to verify the performance of the PAM system.
(6) In Figures 10 and 11, why does the system only provide a rectangular effective imaging area instead of a square imaging area?
(7) The authors demonstrate that "This research introduces a portable PAM system." What are the specific dimensions of the system?
Reviewer 4 Report
Dear Authors,
The proposed paper entitled “Modeling of Rapid Pam Systems Based on Electrothermal Micromirror for High-Resolution Facial Angiography” deals with a portable photoacoustic microscopy system based on a large stroke electrothermal micromirror. The paper focuses more particularly on modelling aspects of the electrothermal actuators. Numerical results about the steady state and the transient responses are presented and an imaging methodology is briefly discussed.
The paper is well and clearly presented but remains rather superficial. The design and the FEM modelling are proposed without any discussion. There is no optimisation procedure, no discussion of expected performance and the influences of different parameters. In my opinion, this is very much missing from this work, especially as there are no experimental results to give more consistency to the whole. The major part of the paper is devoted to the electrothermal actuators and in this context, it is difficult to see a contribution compared to the much more comprehensive reference [28] in terms of modelling, reflection on parameters and experimental results.
The main interest of the article lies in the application to photoacoustic imaging, but the developments around this application are too superficial and would deserve special attention.
For these reasons, I consider that the paper needs to be revised (major revisions). In parallel to the need to amend the article in substance, I am submitting various comments and typographical errors for correction.
- p.2-l.63-64: It is written “… a large braking range …”, I suppose that the authors wanted to write “… a large displacement range …” or “… a large stroke” ?
- p.2-l.72: “… angel.” à “… angle.”
p.2-l.74: “… the reflection … of the micromirror.” à “… the reflectivity (or reflectance) … of the micromirror.”, isn’t it ?
- In Figure 1: “Ultransonic waves” à “Ultrasonic waves”
- p.3-l.106: “… A layer of …” à “… a layer of …”
- p.3-l.107: “The electrothermal actuators … is shown …” à “The electrothermal actuators … are shown …”
- p.3-4: The technological arguments for the choice of the octagonal shape for the “Z” type actuator are not convincing. Moreover, the octagonal shape is not chosen for the “O” type actuator. It would have been more logical to consider a discoidal shape for all types of actuators. What do the authors think about these comments?
- p.4-l.131-134: The authors give the equations (1), (2) and (3) without actually using them. I think that either their use should be developed a bit more or they should be deleted, keeping only the sentence: “The convection heat transfer coefficient hL is set at 1750W/(m2K ) according to the classical Morgan relation [31].”
- p.4-l.140-14, p.5-l.148: Errors about figure references have to be corrected.
- p.4-l.141-142: As previously reported, I don’t understand the term “brake” in this context.
- p.5-l.147: The term “ISC” appears for the first time and has to be explained.
- P.5-6: Figure 4 needs to be improved. The architecture of the structures and in particular the stacking of the different layers is very difficult or impossible to understand. For this, reference should be made to reference [28]. In addition, the values shown on the displacements on the right-hand side of the figure are almost unreadable.
- p.6-l.172-174: The authors said that a prestress of 400MPa is applied to the actuators: does it represent the residual stresses due to the fabrication process (i.e., the releasing of the structure)? And to what extent do the authors say that this protects the structure? And cannot this stress level be detrimental to the integrity of the structure?
- p.6-l.174 and l.178 (maybe in other place of the text): The authors use the term “motivation” for “excitation”, isn’t it? … to be corrected.
- p.8: In Figure 7, the scale of the rotation angle axis could be more appropriate while maintaining the readability of the displacement and rotation curves.
- p.8: It is quite surprising that the rise and fall times of displacements and rotations are exactly the same for “O” and “Z” type actuators. Wouldn't there be an error in reporting the determined values?
Moreover, how these values and in particular the rise times determined? Usually, the time between 10% and 90% of the steady values is considered but it seems that it is not the case in particular for the case (a) (“O” type actuator).
- p. 9: I do not understand the frequency analysis. What is the correspondence between the 40Hz / 0.17Hz excitation and the resulting frequencies of the “O” and “Z” type actuators? Obtaining the double frequency (considering that heating is independent of the voltage sign) seems to be relevant but 80Hz (OK) and 120Hz (???) / 0.18Hz (???) and 0.4Hz (» OK) are not all consistent …
Moreover, in Figure 9a, the point 3 related to the low shaft seems to be characterized by a frequency about 1Hz and not 0.18Hz??? Where is the problem? There is no proposal to phase shift the pairwise excitations. What is the literature on this type of device?

Round 2
Reviewer 3 Report
All my questions are well addressed. It can be accepted now.